# PET-CT in Clinical Adult Oncology: II. Primary Thoracic and Breast Malignancies

**DOI:** 10.3390/cancers14112689

**Published:** 2022-05-29

**Authors:** Matthew F. Covington, Bhasker R. Koppula, Gabriel C. Fine, Ahmed Ebada Salem, Richard H. Wiggins, John M. Hoffman, Kathryn A. Morton

**Affiliations:** 1Department of Radiology and Imaging Sciences, University of Utah, Salt Lake City, UT 84132, USA; matthew.covington@hsc.utah.edu (M.F.C.); bhasker.koppula@hsc.utah.edu (B.R.K.); gabriel.fine@hsc.utah.edu (G.C.F.); ahmed.salem@utah.edu (A.E.S.); richard.wiggins@hsc.utah.edu (R.H.W.); john.hoffman@hci.utah.edu (J.M.H.); 2Department of Radiodiagnosis and Intervention, Faculty of Medicine, Alexandria University, Alexandria 21526, Egypt; 3Intermountain Healthcare Hospitals, Summit Physician Specialists, Murray, UT 84123, USA

**Keywords:** FDG, PET, FES, non-small cell lung cancer, small cell lung cancer, malignant pleural mesothelioma, thymoma, thymic carcinoma, breast cancer, infiltrative ductal carcinoma, invasive lobular carcinoma

## Abstract

**Simple Summary:**

Positron emission tomography (PET), typically combined with computed tomography (CT), has become a critical advanced imaging technique in oncology. With PET-CT, a radioactive molecule (radiotracer) is injected in the bloodstream and localizes to sites of tumor because of specific cellular features of the tumor that accumulate the targeting radiotracer. The CT scan, performed at the same time, provides information to facilitate assessment of the amount of radioactivity from deep or dense structures, and to provide detailed anatomic information. PET-CT has a variety of applications in oncology, including staging, therapeutic response assessment, restaging, and surveillance. This series of six review articles provides an overview of the value, applications, and imaging and interpretive strategies of PET-CT in the more common adult malignancies. The second article in this series addresses the use of PET-CT in breast cancer and other primary thoracic malignancies.

**Abstract:**

Positron emission tomography combined with x-ray computed tomography (PET-CT) is an advanced imaging modality with oncologic applications that include staging, therapy assessment, restaging, and surveillance. This six-part series of review articles provides practical information to providers and imaging professionals regarding the best use of PET-CT for the more common adult malignancies. The second article of this series addresses primary thoracic malignancy and breast cancer. For primary thoracic malignancy, the focus will be on lung cancer, malignant pleural mesothelioma, thymoma, and thymic carcinoma, with an emphasis on the use of FDG PET-CT. For breast cancer, the various histologic subtypes will be addressed, and will include ^18^F fluorodeoxyglucose (FDG), recently Food and Drug Administration (FDA)-approved ^18^F-fluoroestradiol (FES), and ^18^F sodium fluoride (NaF). The pitfalls and nuances of PET-CT in breast and primary thoracic malignancies and the imaging features that distinguish between subcategories of these tumors are addressed. This review will serve as a resource for the appropriate roles and limitations of PET-CT in the clinical management of patients with breast and primary thoracic malignancies for healthcare professionals caring for adult patients with these cancers. It also serves as a practical guide for imaging providers, including radiologists, nuclear medicine physicians, and their trainees.

## 1. Introduction

PET-CT is an invaluable advanced diagnostic imaging modality in oncology with a variety of applications, including initial staging of cancer, assessment of response to therapy, restaging, and longitudinal surveillance for recurrence. The goal of the second in this six-part series of review articles is to provide practical information for referring providers, radiologists, nuclear medicine practitioners, and their trainees regarding the best use of PET-CT for specific primary thoracic malignancy and for breast cancer, and the potential pitfalls and nuances that characterize these applications. This article also focuses primarily on FDA-approved and clinically accessible radiopharmaceuticals. For primary thoracic malignancy, the focus will be on lung cancer, malignant pleural mesothelioma, thymoma, and thymic carcinoma, with an emphasis on the use of ^18^F fluorodeoxyglucose (FDG) PET-CT. For breast cancer, various histologic subtypes will be addressed, including invasive ductal carcinoma, invasive lobular carcinoma, and other malignant, atypical, and benign breast lesions. The discussion will include not only FDG but also the use of two other FDA-approved agents, ^18^F-fluoroestradiol (FES), which binds to estrogen receptors (ERs) for assessment of ER+ breast malignancy, and ^18^F-sodium fluoride (NaF) for bone imaging. Because PET-MR scanners are in limited use, the focus here is on PET-CT because if its widespread availability. However, basic principles described are also applicable to PET-MR. The targeted readers for this review are imaging providers, including radiologists, nuclear medicine physicians, and their trainees. Interpretive guidance is provided to these providers. The information is also important to medical and surgical care professionals caring for or treating adult cancer patients.

## 2. Lung Cancer

Lung cancer is one of the most frequently diagnosed malignancies and the leading cause of cancer-related mortality worldwide in both men and women, with an estimated 2.1 million new cases and 1.8 million deaths per year [1]. The most important risk factor for developing lung cancer is a history of cigarette smoking, with an estimated likelihood ratio estimated at 10:1 between smokers and non-smokers [2]. Other risk factors for lung cancer include occupational exposure to carcinogenic substances such as asbestos, arsenic, beryllium, and other agents, prior history of exposure to radiation, a family history of lung cancer, and increasing age. Lung cancer my present with symptoms or be found incidentally on chest imaging including plain chest radiographs or chest CT. The most common symptoms at presentation include worsening cough, chest pain, hemoptysis, weight loss, shortness of breath, or hoarseness of voice. Additional symptoms may result from local invasion or compression of adjacent thoracic structures such as thoracic pain, compression of esophagus leading to dysphagia, or compression of the superior vena cava causing facial edema and distension of the superficial veins of the head and neck. Symptoms from distant metastases including neurological deficits or personality change resulting from brain metastases or pain due to bone metastases. Infrequently, patients may present with symptoms and signs of paraneoplastic diseases such as hypertrophic osteoarthropathy with digital clubbing, or hypercalcemia from production of parathyroid hormone-related protein by the tumor. Physical examination may identify enlarged supraclavicular lymphadenopathy, pleural effusion or lobar collapse, unresolved pneumonia, or worsening signs of associated disease, such as chronic obstructive pulmonary disease or pulmonary fibrosis. The other primary thoracic malignancies include lymphoma, primary pleural mesothelioma, thymoma/thymic carcinoma, or other tumors arising from vascular structures and heart within the mediastinum, and tumors arising from nervous system.

### 2.1. Normal Physiologic and Benign Patterns

The assessment of patients with suspected intrathoracic malignancy has routinely included morphological imaging evaluation, with either chest X-ray or CT, in conjunction with CT of the abdomen and pelvis, as well as with MRI brain and bone scintigraphy, if indicated. However, integrated FDG PET-CT has now become a preferred imaging modality in the evaluation of multiple cancers, including lung cancer. The anatomic data obtained using CT can be supplemented by the metabolic data obtained by positron emission tomography (PET), and thus the use of hybrid FDG PET-CT provides more diagnostic accuracy than with either modality alone. 

A variety of non-malignant processes can pose challenges in interpretation of PET-CT studies, leading to false-positive findings mimicking malignancy. These findings can be due to physiologic variants, inflammatory processes, infection, iatrogenic conditions, post-procedural/post-therapeutic changes, and benign tumors arising from different thoracic structures. A knowledge of these physiologic variants and false positive findings can reduce unnecessary invasive procedures, such as biopsy and thoracotomy, in patients with benign disease [3]. Physiological variants such as muscle uptake, specifically if involving the chest wall musculature including intercostal muscles, can be mistaken for pleural pathology. Uptake can be seen within the normal thymic tissue, specifically in young patients or in those with post-chemotherapy “thymic rebound”. However, the uptake within the normal or rebound thymic tissue is homogeneous and classically presents as an inverted “V” pattern in the anterior mediastinum in the expected location of the thymus [4]. FDG uptake by hypermetabolic brown adipose tissue (“brown fat” is a well-known cause of false positive finding on PET-CT due to its thermogenic properties and hypermetabolic nature [5] (Figure 1). Frequently, brown adipose tissue uptake is bilateral and symmetric and correlates with areas of slightly increased fat tissue attenuation. The most common location for brown fat uptake of FDG is the neck primarily involving the sub-occipital, cervical, supraclavicular regions, posterior paraspinal/intercostal regions, within the mediastinum, and in the retroperitoneum in perinephric space, including around the adrenal glands. This pattern of FDG activity, especially if asymmetric and if identified within the supraclavicular regions, mediastinum, and along the internal mammary vessels, can mimic malignant/metastatic processes. Lipomatous hypertrophy of the interatrial septum (LHIS) of the heart is characterized by fat accumulation in the interatrial septum with a prevalence of 2–3% which can demonstrate significant accumulation of FDG due to its high concentration of brown fat (Figure 1).

Chronic or acute inflammatory and granulomatous processes such as sarcoidosis are well-known causes of hypermetabolic lymphadenopathy within the mediastinum and hilar regions. Prior granulomatous disease is often associated with calcifications within the nodes and with associated pulmonary nodules, some of which may be calcified. Within the mediastinum and hila, granulomatous disease typically assumes a balanced, symmetrical distribution, whereas adenopathy due to lung cancer is typically more ipsilateral to the side of the primary tumor (Figure 2). However, histologic evaluation could be necessary in differentiating this systemic inflammatory process from advanced neoplastic disease with bilateral mediastinal or hilar hypermetabolic lymphadenopathy. Reactive processes to foreign material within the lung parenchyma or pleura, such as silica or talc, or within the nodes in the hilar region or mediastinum, can mimic malignant processes (Figure 3) [6]. Similar false positive findings can be encountered with acute infective processes such as Coccidioidomycosis as well as various mycobacterial infections, both of which present with similar pulmonary manifestations. The acute phase of these infections and other viral, bacterial, or fungal pulmonary infections demonstrate hypermetabolic parenchymal or nodal lesions on FDG PET-CT and may mimic primary or metastatic lung malignancy (Figure 4) [7]. 

Immunotherapeutic approaches are becoming more common in the treatment of lung cancer, and include a variety of check-point inhibitors, such as PD-1 inhibitors (nivolumab, pembrolizumab, cemiplimab), PD-L1 inhibitors (atezolizumab, durvalumab), and the CTLA-4 inhibitor ipilimumab. These drugs may cause systemic inflammatory reactions, called immune-related adverse events (irAEs) with a variety of manifestations involving many organs and lymph nodes, including sarcoidosis-like reactions, and include a variety of imaging manifestations [8]. These inflammatory responses can mimic disease progression on FDG PET-CT [9]. In many cases, the immunotherapy, can be temporarily discontinued if the irAEs are producing clinically significant complications [10]. In these cases, an FDG PET-CT scan can be considering during the interval of discontinuation to differentiate irAEs from true progression. However, in some cases, a biopsy of target lesions identified by FDG PET-CT may be required to differentiate between irAEs and progressive cancer. 

Post radiation changes present with findings involving anatomic structures with a geographically distinct margins (Figure 5). Accurate clinical history with awareness of the prior radiation field location is essential to avoid misdiagnoses. Procedures such as talc pleurodesis used for treatment of recurrent pleural effusions or pneumothorax can lead to chronic granulomatous inflammatory process with nodularity and associated calcifications along the pleural surfaces and can mimic malignant process [11]. This is typically strongly FDG-avid due to host reactive change. FDG embolus, a tiny hypermetabolic clot within the lung parenchyma without underlying CT correlate, is associated with technique during injection of tracer activity. Follow-up chest CT or attention to this specific finding on future studies would often resolve the diagnostic dilemma associated with this situation [12]. Iatrogenic or traumatic injury to the bones including procedures such as sternotomy or thoracotomy can present with increased FDG uptake at these sites due to the healing process. These conditions can be differentiated from malignancy in that the metabolic activity at these locations is shown to be lower than pathological fractures resulting from malignant processes, which has been found to be statistically significant. However, accurate clinical and surgical history, and careful attention to CT imaging is essential in evaluating FDG avid osseous lesions [13]. 

A myriad of benign tumors involving the chest can demonstrate focal FDG activity. These tumors include nerve sheath tumors, such as schwannomas, osteoblastoma, chondromyxoid fibromas, aneurysmal bone cysts, and brown tumors, among others. These lesions often cannot be differentiated from malignant lesions and additional imaging methods such as CT/MRI may be required to further characterize and, in most cases, tissue sampling may be required. Benign soft tissue tumors such as elastofibroma dorsi can demonstrate focal increased metabolic activity on FDG PET-CT studies [14]. These tumors are often seen in older women with a CT prevalence rate of 2%. These tumors are often bilateral though they can be unilateral and are characterized by fibroblast proliferation and accumulation of elastic fibers. On CT, these tumors demonstrate a combination of muscle and fat tissue attenuation. These tumors, which can be unilateral or bilateral, are typically located along the undersurface of the tips of the scapulae between the ribs and serratus and latissimus dorsi muscles (Figure 6).

### 2.2. Evaluation of Solitary Pulmonary Nodules

A solitary pulmonary nodule (SPN) is defined as a single round/oval lesion which is less than 3 cm in size surrounded by lung parenchyma and without associated collapse of the distal lung or lymphadenopathy [15]. Early-stage lung cancer presents as a solitary pulmonary nodule in many patients. However, SPNs are very common, affecting up to 0.1–0.2% of the adult population, and are more common in areas of endemic fungal disease or in older patients. SPNs can be detected incidentally or as a part of screening with low-dose chest computed tomography (LDCT). It is important to establish the nature of SPNs and determine whether they are benign or malignant, though this is not always feasible with conventional CT alone. LDCT nodule screening has been shown to reduce mortality resulting from lung cancer by 20% [16]. Currently, annual screening for lung cancer is recommended with LDCT in adults aged 55–80 years with a 30-year history of smoking who currently smoke or have quit smoking within the last 15 years [17,18]. However, 96% of the nodules detected on screening LDCT are non-malignant and any further evaluation of these nodules adds to the cost and procedure-related complications [19].

When a solitary pulmonary nodule is identified on a chest radiograph or chest CT, prior available imaging must be reviewed. If the nodule were present at an earlier time, then the doubling time of the lesion must be assessed. If the lesion has not grown over a period of 2 years, then the lesion is most likely benign, especially in the absence of CT findings suggestive of malignancy such as spiculated margins, location within the lung parenchyma, and the clinical parameters of the patients, such as age, smoking history, or other history of cancers [20]. 

FDG PET-CT is shown to be useful in the assessment of solitary pulmonary nodules >8 mm in diameter and is advised in patients who are at low or moderate risk for malignancy (5–20% and 20–80% of all patients with an SPN, respectively), and depending on their radiological characteristics. In addition, FDG PET-CT could benefit patients at high risk for SPN malignancy, by defining the local extent of the lesion and identified regional nodal spread or distant metastases [21]. CT is an excellent imaging modality in the evaluation of SPN, due to its very high sensitivity. However, the specificity of CT is limited. The introduction of FDG PET-CT imaging into clinical practice has had a positive impact on diagnostic specificity in patients presenting with SPNs, because it provides functional as well as anatomical data [22]. FDG PET-CT parameters that are considered suspicious for include a SUVmax of >2.5, metabolic activity greater than that of mediastinal blood pool, or any visually appreciable metabolic activity greater than background lung parenchyma for a lesion <1 cm in diameter [23]. These parameters have the same likelihood of malignancy as recognition of growth of a nodule in a time frame suspicious for lung cancer on CT. 

A number of pitfalls can be a hindrance in the assessment of SPNs with FDG PET-CT including inflammatory conditions or infections such as bacterial or fungal infections; and granulomatous entities such as tuberculosis, sarcoidosis, and histoplasmosis, all of which exhibit increased metabolic activity [12,24]. Additionally, FDG PET-CT can result in false-negative studies due to limited spatial resolution of PET cameras, resulting in partial volume effects that limit characterization of very small nodules. Additional causes of false-negative scans are lepidic-type adenocarcinoma (which may be low in metabolic activity), typical bronchial carcinoid tumors, and metastatic disease from tumors that may be low in metabolic activity (such as renal cell carcinoma, some lower grade sarcomas, and some testicular cancers (Figure 7). As such, a lung nodule that is not hypermetabolic on FDG PET-CT must nonetheless be followed by CT according to conventional Fleischner criteria.

### 2.3. Non-Small Cell Lung Cancer

According to World Health Organization classification, lung cancer is broadly categorized into non-small cell lung cancer and small cell lung cancer. Non-small cell lung cancer (NSCLC) is the predominant histological variant, accounting for 85–90% of all cases of lung cancer and encompasses three subtypes: squamous cell carcinoma (25%), adenocarcinoma (40%), and large cell carcinoma (10%). The remaining 10–15% of cases represent small cell lung cancer (SCLC). The incidence of squamous cell lung cancer strongly associated with cigarette smoking has decreased over the last few years, whereas adenocarcinoma has become the predominant histological subtype of lung cancer and can also be found in non-smokers. FDG PET-CT has become the mainstay imaging modality with significant impact in the management of patients with NSCLC. A meta-analysis has concluded that FDG PET-CT can diagnose malignant pulmonary lesions with an estimated sensitivity of 96.8% and specificity of 77.8% [25]. Another study by Budak et al. demonstrated a sensitivity of 94% for FDG PET-CT in detecting malignancy, and additionally demonstrated a change in treatment plan in 34% of patients based on PET-CT findings [26]. 

In patients with newly diagnosed NSCLC, the initial disease staging is a crucial step in selecting the most appropriate therapy and determining the prognosis [15]. The International Association Society of Lung Cancer (IASLC) recommends the TNM classification of malignant tumors published by the International Union Against Cancer (IUAC) and the American Joint Committee on Cancer (AJCC), and the latest eighth edition was published in 2016, which is the established, uniform method of staging lung cancer and depends primarily on the anatomic extent of disease [26]. The primary tumor (T), regional lymph nodal involvement (N), and distant metastases (M) are the basis for initial lung cancer screening. Based on the different T, N, and M descriptor combinations, patients are grouped into different stages, which determine the clinical management and can also predict various prognoses. TNM staging system can be used for staging both NSCLC as well as SCLC. An appropriate differentiation between patients with potentially curable disease and those who will be receiving palliative therapies is of utmost importance. It has been shown in multiple studies that integrated FDG PET-CT is more accurate in characterizing TNM status than CT alone, PET alone, or visual correlation between separately acquired PET and CT scans [12,24,27].

In Refs. [15,28], T staging describes the location, size, and extent of the primary tumor, as well as the presence of absence of satellite nodules. Given its excellent anatomical resolution, CT remains an important modality for T stage assessment although its ability to evaluate soft tissue invasion and its ability to distinguish primary lesions from post-obstructive atelectasis is limited. FDG PET-CT significantly improves T staging, due to its precise CT correlation with the degree of FDG uptake (Figure 8). Multiple studies have shown that integrated PET-CT provides crucial information on mediastinal infiltration and chest wall invasion, and aids in differentiating between tumor and post-obstructive atelectasis [2,29]. 

N staging identifies nodal involvement in lung cancer is of paramount importance, especially in patients with mediastinal disease without distant extra-thoracic disease. In these patients, the N stage will have therapeutic and prognostic implications [29]. In describing areas of nodal involvement, it is important to identify the specific nodal location involved, as outlined by the IASCL, as this will affect the N classification [26]. Patients staged as N0 or N1, without lymph node involvement, are generally treated with local intervention, and patients with N2 disease with ipsilateral mediastinal lymph node metastases might benefit from a combined approach with local and systemic therapies. N3 diseases with contralateral mediastinal lymph node metastases are considered incurable and will eventually require palliative care [2]. FDG PET-CT has advantages over CT alone, even when the lymph nodes are smaller than 10 mm in diameter [2,30]. FDG PET-CT is of crucial importance in evaluating nodal sites that are inaccessible to mediastinoscopy, such as the aortopulmonary window, anterior mediastinum, and posterior subcarinal nodes. Even though FDG PET-CT is an excellent noninvasive imaging modality in the detection of nodal metastatic involvement, mediastinoscopy remains the gold-standard and needs to be performed wherever there is ambiguity or uncertainty with regard to the status of any one lymph node in patients with NSCLC (Figure 9).

Nearly half of all patients with NSCLC have distant metastatic disease at initial diagnosis and identification of distant metastases is of major importance in the management and prognosis. Additionally, among patients who have been treated with radical and supposedly curative therapy, approximately 20% are likely to develop recurrent disease due to undetected foci of metastasis at initial M staging. Distant metastases are most commonly seen involving the brain, skeleton, liver, and adrenal glands in descending order of frequency (Figure 10) [31]. Traditional evaluation for distant metastatic disease includes CT scans of chest, abdomen, and pelvis, brain imaging with CT or MRI and bone scintigraphy [2]. However, FDG PET-CT has been shown to be great utility in M staging of the patients with NSCLC, specifically if the patients present with clinical manifestations of metastatic disease. Additionally, FDG PET-CT has been shown to provide more pertinent information during the preoperative assessment than is CT alone, except in the assessment of brain metastases, in a situation where the two modalities yield similar results [32]. For staging mediastinal lymph nodes, diffusion-weighted imaging MRI (DWI) and FDG PET-CT showed similar performance in staging of mediastinal lymph nodes, supporting the idea that DWI may offer an alternative to FDG PET-CT in some cases [33].

Adrenal masses can be present in up to 20% of patients with NSCLC and approximately two thirds of these lesions are adrenal adenomas. FDG PET-CT also has high sensitivity and specificity in evaluating adrenal masses, and helps in avoiding unnecessary procedures. False-negative result from partial volume effect in very small (<1 cm) lesions, or due to hemorrhage, or micro-metastases in which FDG uptake can be low. Benign adrenal adenomas can occasionally display high levels of FDG uptake, leading to false-positive results. Histological confirmation is recommended in such situations when management depends solely on characterization of adrenal lesions. FDG PET-CT is also useful in identifying metastases in uncommon sites that would otherwise escape detection, such as within soft tissue structures or small lymph nodes in the supraclavicular region or within the retroperitoneum, with the odds of identifying these sites of disease being 5–29% [32]. 

FDG PET-CT is helpful in guiding biopsy procedures with accurate needle placement within the viable portion of the lesion, thus significantly increasing the chances of achieving a definitive histologic diagnosis, contributing substantially to the management and treatment of lung cancer [34]. Up to 75% of all NSCLC patients could benefit from radiotherapy at some point during their treatment [35,36]. The higher diagnostic accuracy of FDG PET-CT has been shown to have a significant impact on radiotherapy planning for lung cancer as it delineates the extent of tumor(s) with precision, thus preventing mistargeting and unnecessary irradiation of adjacent structures such as esophagus and other mediastinal structures [37]. Higher sensitivity of FDG PET-CT in detecting abnormal mediastinal lymph nodes allows FDG PET-CT to be used in performing target volume depiction using anatomical biological contouring, with much better results [2].

### 2.4. Small Cell Lung Cancer

Small cell lung cancer (SCLC) is an extremely aggressive tumor type which accounts for about 10–15% of all lung cancer cases [38,39]. It originates from neuroendocrine precursor cells and is characterized by rapid growth and early metastatic potential, with approximately 70% of patients presenting initially with metastases [40]. Up to 10–25% of patients with SCLC have brain metastases at the time of initial diagnosis and an additional 40–50% will develop brain metastases during the course of their disease. Although SCLC is usually chemo-sensitive in the early stages and sensitive to radiation therapy, most patients with SCLC will experience recurrent disease leading to death. Over 95% of patients with SCLC are either current or former smokers [2]. According to the Veterans Administration Lung Group’s criteria, SCLC has been traditionally divided into two stages—a binary system of staging—limited disease and extensive disease [41]. Limited-stage disease SCLC (LD-SCLC) is disease confined to ipsilateral hemithorax, mediastinum (including contralateral mediastinal nodes), and ipsilateral supraclavicular nodal involvement, which can be encompassed within a single radiation field (Figure 11) [42]. Extensive-stage disease SCLC (ED-SCLC) is spread beyond the ipsilateral hemithorax including hematogenous metastatic disease and presence of malignant pleural or pericardial effusion (Figure 12) [39,43]. Standard therapy for patients with LD-SCLC includes chemotherapy in conjunction with radiation therapy. Due to high potential for developing brain metastases in this group of patients, palliative cranial radiation is also indicated to increase overall survival. In ED-SCLC, only systemic chemotherapy is considered, as a palliative treatment, because long-term survival in these patients is rare [43,44]. According to the more recently adopted AJCC (American Joint Commission on Cancer) TNM staging system 8th edition, there is no significant difference between NSCLC and SCLC in the staging system [26]. The NCCN Panel adopted a combined approach for staging SCLC using both the AJCC TNM staging system and the VA scheme for SCLC. In applying the TNM classification to the VA system, the so-called limited-stage SCLC is defined as stage I–III (any T, any N, and M0, respectively), which can be effectively treated with definitive radiation therapy and extensive-stage SCLC is defined as stage IV (any T, any N and M1a/b) or T3–T4, harboring multiple lung nodules or tumor volume that is too large to be encompassed in a tolerable radiation plan/field. 

A number of diagnostic modalities have been applied to the staging of SCLC. CT of the chest, CT of the abdomen/pelvis, and MRI of the brain are the most common for staging SCLC, with inclusion of Tc-99m bisphosphonate bone scan in some cases. Multiple studies have evaluated the diagnostic accuracy of FDG PET-CT staging compared to conventional staging in patients with SCLC [45,46,47]. In a prospective study by Fischer et al., the diagnostic performance of FDG PET-CT in correctly diagnosing LD-SCLC and ED-SCLC was 95% and it was higher compared with that of conventional imaging methods with diagnostic accuracy of 85% [47]. The better diagnostic performance of FDG PET-CT in correctly staging LD-SCLC and ED-SCLC was also confirmed by another prospective study by Kishida et al., who found a diagnostic accuracy of 96.6% for FDG PET-CT compared to 91.5% correctly staged as ED-SCLC or LD-SCLC for conventional staging, although this difference was not statistically significant [45]. Few studies have also assessed the change of binary SCLC staging using FDG PET-CT compared to conventional staging (upstaging from LD-SCLC to ED-SCLC or downstaging from ED-SCLC to LD-SCLC), reporting percentages ranging from 5.1 to 21.7% [45]. A recent report supported that whole-body MRI and FDG PET-MRI may offer some advantages over FDG PET-CT and conventional CT in T-staging of the tumor, particularly in determining whether local invasion is present [48].

In a study by Lee et al., FDG PET-CT and bone scintigraphy were compared for SCLC staging based on bone metastases. The sensitivity of FDG PET-CT was 100% and 87% compared with the sensitivity of bone scintigraphy with 37% and 29% on a per-patient- and on a per-lesion-based analysis, respectively [49]. On the other hand, FDG PET-CT cannot substitute for brain MRI due to the limited detection rate of brain metastases by FDG PET-CT because of the high normal background FDG uptake in normal brain [45].

PET-CT has been shown to modify therapeutic interventions in up to 10–33% of patients based on the PET-CT findings [29]. Patients could benefit from correct upstaging from limited disease to extensive disease, reducing the ineffective approaches with their associated morbidity. Similarly, based on PET-CT findings, patients with indeterminate M stage could be correctly down-staged from extensive disease to limited disease and potentially offered curative treatments. As in patients with NSCLC, the targeted volume delineation for the purpose of radiotherapy planning for SCLC could be more precisely outlined by PET-CT [29].

In summary, PET-CT assessment of treatment response has been proven to be useful in SCLC, as most tumors will undergo metabolic changes before morphological changes in involved structures occur. PET-CT can detect early recurrent disease, as well as residual disease in treated tissue. It has also been demonstrated that there is a positive relationship between SUVmax values and overall survival rates in SCLC [2]. Since the natural history of SCLC indicates an unfavorable prognosis, early detection of a response to treatment is important and can be elucidated by FDG PET-CT [41].

## 3. Malignant Pleural Mesothelioma

Malignant pleural mesothelioma (MPM) is an aggressive tumor associated with a prior history of asbestos exposure in 70% of cases. There has been a declining incidence of MPM over the past 15 years due to restrictions in asbestos use. However, because sporadic cases due occur and the latency following asbestos exposure is approximately 40 years, it continues to be a tumor that is not infrequently encountered in clinical practice. There is also a higher incidence of MPM in patients previously having received thoracic therapeutic radiation. MPM is a malignancy that arises from the serosal surface lining the pleura, although extra-pleural mesothelioma of the peritoneum and pericardium can also occur. Three histologic variants of MPM exist, including epithelioid (the most common and most favorable), sarcomatoid, and biphasic. The median age is 70–72 years [50]. There are competing staging systems for MPM with the traditional TNM system remaining most common. Diagnosis is typically made by pleural fluid cytology or fine needle biopsy. Aggressive surgical removal (pleurectomy, decortication, pneumonectomy) is preferred if the patient is deemed operable. Combined modality treatment is often employed. Despite aggressive treatment, survival remains relatively poor, with all-stage 5-year survival of 10% [50]. 

In diagnostic imaging of MPM, FDG PET-CT, whole-body MRI and PET-MRI show superiority over conventional CT [51]. MPM is typically intensely hypermetabolic on FDG PET-CT, making it highly amendable to staging by this method (Figure 13 and Figure 14). Hypermetabolic pleural thickening is typically associated with calcified pleural plaques but can be mimicked by hypermetabolic host inflammatory reaction to talc pleurodesis, empyema, and other pleural inflammatory disease, or by other subpleural malignant processes (Figure 15) [52]. NCCN endorses the use of FDG PET-CT to stage patients only if they are being considered for surgery [53]. However, FDG PET-CT may under-stage mediastinal or nodal disease in MPM and is typically best used for assessment of distant metastases [54]. PET-CT should be done prior to contemplated talc pleurodesis. NCCN also supports the use of FDG PET-CT for mediastinal assessment based on possible evidence of disease progression following induction chemotherapy.

In summary, MPM is typically intensely hypermetabolic on FDG PET-CT, but can be mimicked by hypermetabolic host inflammatory reaction to talc pleurodesis, or other inflammatory or malignant pleural processes. The best use of FDG PET-CT for staging should be reserved for patients considered for surgery or for assessment of suspected disease progression despite induction chemotherapy, with the realization that FDG PET-CT may under-stage mediastinal or nodal disease. 

## 4. Primary Thymic Tumors

Primary thymic tumors consist of thymomas, thymic carcinomas, and thymic neuroendocrine tumors. The peak age for these thymic tumors is 70–74 years, with an overall incident of approximately 1.06 per 100,000 population [55]. All arise from the epithelial cells of the thymus. Each type exists as a complex categorization of subtypes, and a continuum from benign to aggressive behavior and poorer prognosis, with a 10-year disease-free survival ranging from 100% for medullary thymoma to 28% for higher grade thymic carcinoma [56]. Thymoma and thymic carcinoma can present with hoarseness due to recurrent laryngeal nerve involvement, compression with superior vena cava syndrome, chest pain, cough, or dysphagia. Thymoma accounts for 20–25% of tumors of the mediastinum and 50% of those in the anterior mediastinum, with most patients presenting in their 40s and 50s. Myasthenia gravis occurs in up to 50% of patients with thymoma although only 10–15% of patients with myasthenia gravis have a thymoma. Thymoma is also associated with a variety of hematologic disorders (such as pure red cell aplasia) and other neuromuscular syndromes. The hematologic manifestations of thymoma will often regress with surgical removal of the thymoma but often recur later, even without recurrence of the thymoma.

Anterior mediastinal masses present a challenge to both conventional imaging and FDG PET-CT. There is a relatively large variation in the size of the thymus in normal individuals. In those younger in age, and those with thymic hyperplasia and thymic stimulation, a mass-like appearance of the thymus may be present and can be moderately hypermetabolic on FDG PET-CT. Many types of tumors may present, or be associated with, anterior mediastinal masses [56]. These include mature and immature teratomas and other germ cell tumors, lymphoma, salivary-type tumors, and metastatic disease. Substernal thyroid tissue can also masquerade as a thymic neoplasm. In general, thymoma tends to be lower in metabolic activity on FDG PET-CT than is thymic carcinoma (Figure 16). However, attempts to utilize FDG PET-CDT to distinguish hyperplastic thymic tissue, benign thymoma, and thymic carcinoma, and the assessment of varying grades of each have been met with variable success [57]. Kumar et al. report the mean metabolic activity (SUVmax) of thymic masses: thymic hyperplasia (SUVmax 1.1), thymoma (SUVmax 2.3), and thymic carcinoma (SUVmax 7.0) [58]. They further reported that an SUVmax of 6.5 could differentiate between thymoma and thymic carcinoma with a sensitivity and specificity of 100% and 87.2% (Figure 16) [59]. Others have reported that FDG PET-CT can distinguish between thymoma and thymic carcinoma but not between thymomas of various grades [60]. However, other reports suggest that metabolic activity alone cannot distinguish between all grades of thymoma [61]. Thymic neuroendocrine tumors tend to be mild in uptake of FDG and may be better imaged by ^68^Ga or ^64^Cu DOTATATE PET [62]. 

In summary, the use of FDG PET-CT in the characterization of anterior mediastinal masses, and to distinguish between various grades of thymoma and thymic carcinoma is an imperfect process. Accurate interpretation requires comparison with prior imaging, clinical information, anatomic imaging characterization, and, ultimately, histological analysis in many cases. In staging thymic carcinoma, the value of FDG PET-CT is still not firmly established [63]. The best use of FDG PET-CT is likely in identifying distant metastases.

## 5. Breast Cancer

### 5.1. Current and Emerging PET Agents and Imaging Technologies 

Breast cancer is currently the most diagnosed cancer worldwide [64]. PET-CT for breast cancer evaluation most commonly utilizes FDG. FDG PET-CT has demonstrated clinical utility for breast cancer staging in cases where locally advanced, metastatic, or recurrent disease is expected, and for imaging evaluation of treatment response [65,66]. Beyond FDG, other U.S. Food and Drug Administration (FDA) approved radiopharmaceuticals for breast cancer imaging, at time of this writing, are ^18^F-fluoroestradiol (FES), which binds to estrogen receptors (ERs) for assessment of ER+ breast malignancy, and ^18^F-sodium fluoride (NaF), that allows assessment of osseous metastatic disease for breast and other cancers [67,68,69]. Of these, FDG and NaF are incorporated into National Comprehensive Cancer Network (NCCN) and European Society for Medical Oncology (ESMO) guidelines for breast cancer management [70,71].

The newest U.S. FDA approved radiopharmaceutical, FES (Cerianna^TM^, Zionexa, Fishers, IN, USA, subsequently acquired by GE Healthcare, Chicago, IL, USA), utilizes binding of radioactive-labeled estradiol to estrogen receptors, thereby allowing direct imaging of estrogen receptor uptake within breast malignancy [72,73,74,75,76,77,78,79,80,81,82]. Prior studies have shown that FES binding to the ER is essentially equivalent to that of endogenous estradiol and FES uptake on PET imaging has been shown to correlate with ER expression on in vitro assays such as radioligand binding or immunohistochemistry [81,82]. FES PET-CT may guide therapeutic decision making by documenting the presence of estrogen receptor heterogeneity in primary and metastatic invasive breast cancer lesions, which can be useful to guide therapeutic decision making between estrogen-targeting hormonal therapies versus chemotherapy [75,83,84]. Additionally, FES-PET uptake at sites devoid of native estrogen receptor expression, such as bone, may be highly specific for breast cancer metastases [83]. 

Beyond the currently FDA-approved radiopharmaceuticals for breast cancer evaluation, early studies have shown utility of other PET imaging agents to depict sites of breast cancer. Among potential emerging PET radiopharmaceuticals for breast cancer evaluation are various types of human epidermal growth factor receptor-2 (HER2)-binding radiopharmaceuticals such as ^89^Zr trastuzumab; progesterone receptor-binding agents, such as ^18^F fluorofuranylnorprogesterone (FFNP); cell proliferation markers such as ^18^F fluorothymidine (FLT); and other radiopharmaceuticals, such as ^18^F-fluorocholine, ^68^Ga PSMA-11, ^68^Ga FAPI, ^89^Z atezolizumab, and ^18^F fluciclovine [66,68,85,86,87,88,89,90,91]. In addition, novel radiochemistry approaches, such as ^18^F-fluoroglycosylation, open the door to the development of a wide variety of novel radiolabeled molecules for use in PET. These show excellent stability and clearance. These can be constructed to target a myriad of cell-specific features important in personalized medicine breast cancer and other malignancies [92].

### 5.2. Whole-Body and Breast-Specific PET Systems

Standard PET imaging for breast cancer evaluation typically denotes use of whole-body PET-CT systems. Alternatively, PET/MRI systems are increasingly being utilized for evaluation of disease extent or treatment response in breast cancer [66,93]. Breast-specific PET imaging using positron emission mammography (PEM) or dedicated breast PET systems (dbPET) have also been proposed for breast cancer imaging [72,94,95,96,97]. These typically offer improved PET resolution compared with whole-body PET cameras, may allow imaging in the prone position similar to breast MRI, and allow PET guidance for tissue biopsy [66,97]. Note that molecular breast imaging and breast-specific gamma imaging typically refer to imaging systems that provide gamma-camera-based imaging utilizing ^99m^Tc-sestamibi. These terms do not apply to breast-specific PET [98].

### 5.3. Indications for Breast FDG PET-CT 

FDG PET-CT imaging for breast cancer is performed to evaluate for locally advanced or distant metastatic disease, therapy response assessment, and evaluation for disease recurrence [66,70,71,91]. Proposed uses for breast-specific PET imaging with PEM or dbPET include local extent of disease evaluation and neoadjuvant therapy response assessment [97]. Unlike mammography, breast ultrasound, and breast MRI, PET imaging is not indicated for primary or supplemental breast cancer screening or for response assessment of primary or axillary lymph node metastatic disease [99]. FDG PET-CT may be beneficial for evaluation of therapeutic response for metastatic breast cancer, primarily through earlier detection of recurrent or progressive disease compared with CT and bone scintigraphy. FDG PET-CT demonstrates low sensitivity for detection of axillary nodal metastatic disease compared with sentinel lymph node biopsy and should not be used in lieu of this procedure [93,100].

### 5.4. Physiologic and Benign Uptake Patterns

FDG PET-CT imaging for breast cancer relies on differences in glucose metabolism to differentiate between normal, benign, and malignant tissues [101]. Physiologic levels of FDG uptake in normal breast tissue normally declines with increasing age and demonstrates higher background uptake in individuals with dense breast tissue [101,102]. A low level of symmetric, diffuse uptake is a normal physiologic pattern in the breast, whereas focal uptake above background of bilateral breast tissue may indicate presence of malignancy [103]. FDG background uptake in the breasts is generally increased among lactating individuals compared with non-lactating individuals (Figure 17) [104]. Fat necrosis is metabolically active and can remain so indefinitely. This is common in the post-operative breast or along the edges of a tram flap reconstruction. Silicone granulomas, either along the edge of a prosthesis or in associated lymph nodes, are typically hypermetabolic on FDG PET-CT (Figure 17). Ultrasound or silicon-specific MRI sequences can clarify these findings as silicone-related (Figure 17). 

FDG normally demonstrates high levels of physiologic uptake in the brain and urinary collecting system and variable levels of uptake in the liver, bowel, and bone marrow, which can reduce sensitivity for evaluation of metastatic at these locations. Conversely, FES typically shows a diffuse, low-level physiologic breast uptake. FES shows high levels of physiologic uptake in the liver, biliary tree, bowel, and urinary collecting system, and variable uptake in the uterus, which can reduce sensitivity for evaluation of metastatic disease at these locations and result in a false-negative disease assessment [93]. If FES-PET/CT is performed while on anti-hormonal therapies such as fulvestrant or tamoxifen, then a false-negative scan may result because FES will be unable to effectively bind to estrogen receptors at sites of malignancy due to binding competition or downregulation of estrogen receptors as a result of therapy. False positive FES uptake is not well established. However, several non-breast malignancies can show FES uptake such as in tumors of the endometrium, uterus, ovaries, and in meningiomas. See Figure 18 for comparison of normal physiologic FDG and FES PET uptake patterns. 

^18^F-NaF shows uptake that predominantly localizes diffusely throughout the bones and focally at sites of degenerative and/or post-traumatic change as well as sites of osseous metastatic disease (Figure 19). NaF PET-CT may occasionally show uptake in primary breast tumors irrespective of frank tumoral calcification, similar to that seen on ^99m^Tc bisphosphate bone scans, but demonstrates an overall low sensitivity for detection of non-osseous sites of breast cancer [105]. Note that incidental focal abnormal FDG uptake in a breast lesion seen on an FDG PET-CT exams performed for indications other than breast cancer has a likelihood of malignancy that ranges from 27 to 55% [103,106,107]. 

### 5.5. Invasive Ductal Carcinoma 

The most common subtype of breast cancer is invasive ductal carcinoma (IDC) which accounts for most invasive breast cancer diagnoses [93,108] Invasive ductal carcinoma characteristically shows abnormal uptake on FDG PET-CT imaging and demonstrates higher uptake levels with larger tumors, higher grade tumors, metaplastic tumors, hormone receptor negative tumors, and triple-negative tumors (Figure 20) [109,110,111]. On FES PET-CT imaging, invasive ductal carcinoma would typically show uptake if the IDC is ER+. FES PET-CT may also depict heterogeneous ER expression in cases of metastatic IDC, wherein some but not all tumors have functional ERs. Invasive ductal carcinoma may be identified at a higher rate on FDG PET-CT imaging compared to invasive lobular carcinoma, discussed further below [110].

### 5.6. Invasive Lobular Carcinoma

Invasive lobular carcinoma (ILC) is a distinct molecular and pathologic entity from invasive ductal carcinoma and is the second most common type of breast cancer after invasive ductal carcinoma, comprising approximately 10–15% of invasive breast cancer cases in the United States [112,113,114,115,116]. Compared with IDC, ILC is more often difficult to identify on mammography, ultrasound, and FDG PET-CT [110,112,113,115,116]. ILC is also more often multifocal and bilateral compared with IDC [117]. Some ILC show hypermetabolic changes on FDG PET-CT as well as uptake on FES PET-CT (Figure 21). ILC may not take up FDG as readily as IDC for reasons that include lower tumor microvascularity, cellular density, proliferation rate, and number of glucose transporters [115,116,118,119,120,121,122,123]. ILC osseous metastases are more frequently present on FDG-PET-CT compared with IDC [118,119] as ILC osseous metastases are more frequently sclerotic, whereas FDG-PET-CT is more sensitive for lytic osseous metastases. Sclerotic ILC osseous metastases also may be indistinguishable from benign bone islands on CT at initial staging, thereby necessitating biopsy or imaging follow-up for confirmation of osseous metastatic disease [118,119]. Metastatic with of ILC may also occur at unusual sites, such as the stomach, uterus or colon [124].

ILC demonstrates higher rates of ER positivity compared to IDC with studies showing greater than 90% positivity for ILC tumors [113,116]. For example, data from the Surveillance, Epidemiology, and End Results Program (SEER) shows ILC demonstrates approximately 95% ER positivity when considering over 17,000 ILC cases [124]. FES-PET may therefore be well-suited to image a high proportion of patients with ILC given the high rate of ER+ disease [125]. 

### 5.7. Other Malignant, Atypical, and Benign Breast Lesions

FDG PET-CT imaging can be performed for staging and evaluation of recurrence for malignant phyllodes tumors of the breast which are locally aggressive tumors with a high rate of local recurrence and risk of hematological spread to extra-nodal sites that include lung, liver, brain, and bone [126] (Figure 22). FDG PET-CT may also have utility for evaluation of other breast malignancies including mucinous carcinomas, papillary carcinoma, medullary carcinoma, ductal carcinoma in situ, and atypical lesions such as atypical ductal hyperplasia [103,110,127]. Papillomas and papillary lesions of the breast, with or without atypia, may show variable abnormal FDG uptake, as can benign entities such as fat necrosis, gynecomastia, fibroadenomas, and silicone granulomas (refer to Figure 17) [103,128]. Primary breast lymphoma [129] (Figure 23) and breast-implant-associated anaplastic large cell lymphoma [130] are also amenable to staging and response assessment evaluation using FDG PET-CT. Although metastatic disease to the breast is rare for most tumors, lymphoma, melanoma, ovarian carcinoma, and lung carcinomas are the most common malignancies to spread to the breast and are typically depicted well on FDG PET-CT imaging [131].

In summary, PET imaging of breast cancer is a rapidly progressing with multiple currently approved and numerous emerging PET radiopharmaceuticals that evaluate various aspects of breast cancer physiology. Whole-body FDG PET-CT remains the primary PET approach for breast cancer staging, evaluation for recurrence, and therapy response assessment. FDG PET-CT may also provide useful evaluation for other types of breast cancer as well as other malignancies that involve the breast such as lymphoma. FES PET-CT allows assessment of sites of ER+ breast cancer as an adjunct to biopsy. 

## 6. Conclusions

The role of PET-CT, and the diversity of radiopharmaceuticals available for tumor imaging, is an ever-evolving field. The degree to which PET-CT contributes to the overall clinical management of patients with cancer is dependent on many important factors that require dedication, ongoing self-education, and effective communication between the providers and imaging professionals. An understanding of the normal biodistribution and physiologic parameters that may alter the distribution of a specific radiopharmaceutical being used, as reviewed in this article for thoracic and breast malignancies, is needed in order to accurately interpret the scans. Pitfalls, artifacts, and normal variants that may simulate disease vary as a function of the radiopharmaceutical used and the organ system examined. A familiarity with multiple factors is required for useful and trustworthy reports, including a knowledge of the specific type of cancer being evaluated, an awareness of the typical spectrum of clinical and imaging findings, the expected patterns of spread of disease, the staging systems specific to the type of tumor being assessed, the therapeutic options, and the typical complications of those therapies. Finally, and perhaps most importantly, it is critical to understand the patient’s history and the specific questions being asked by the referring providers. In evaluating PET-CT scans for thoracic and breast malignancies, the importance of engaging imaging specialists in chest radiology and breast imaging cannot be overstated. If diligence is applied to ensure that all of these requirements are met and maintained, the radiologist or nuclear medicine practitioner becomes a true member of the interdisciplinary healthcare team. In addition, the referring providers should understand the limitations and advantages of PET-CT, communicate well with the imaging professionals, and ask specific questions that will guide the imagers in providing the most relevant and accurate information.

## Figures and Tables

**Figure 1 cancers-14-02689-f001:**
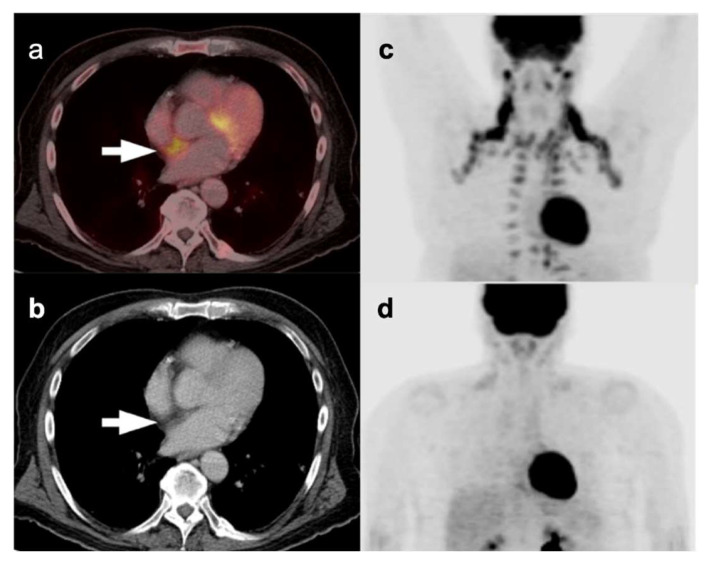
Variable appearance of hypermetabolic brown adipose tissue. (**a**) Axial FDG PET-CT and (**b**) CT images of the chest show typical appearance of lipomatous hypertrophy of the interatrial septum (LHIS) (**white arrows**); (**c**) MIP images of FDG PET scans of a patient who arrived cold for a PET-CT scan and showed hypermetabolic brown adipose tissue in the neck, axillas and paraspinous intercostal regions.; (**d**) A repeat study performed on the same patient as (**c**) with careful attention to maintaining a warm environment for the day prior to the repeat scan and during the interval from injection to imaging (**b**, **lower panel**) shows near complete resolution of hypermetabolic brown adipose tissue.

**Figure 2 cancers-14-02689-f002:**
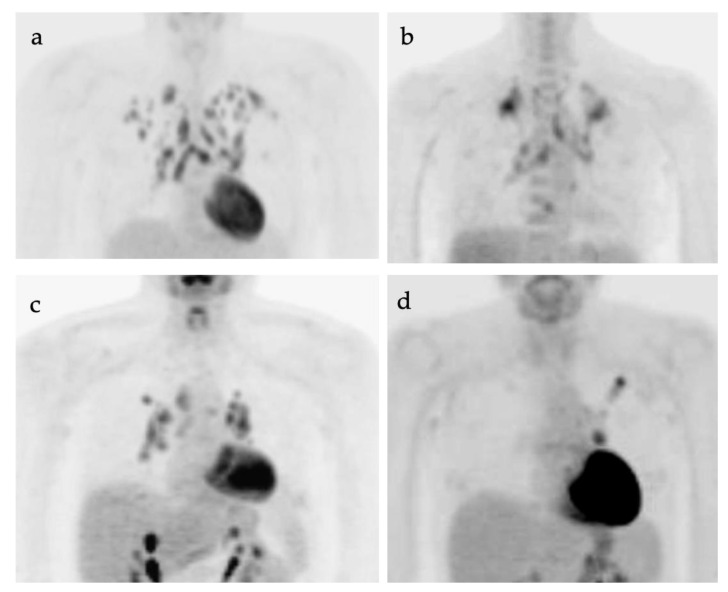
Various patterns of typical lung findings on FDG PET MIP images. (**a**) Multiple predominantly upper lobe hypermetabolic nodules with bilateral hilar, mediastinal, and subcarinal hypermetabolic nodes in a patient with silicosis, but the pattern could also be seen with sarcoidosis; (**b**) bilateral superior hilar retraction and hypermetabolism of hilar lymph nodes and upper lobe masses, in this case due to reactivation tuberculosis; (**c**) bilateral hilar and mediastinal lymph nodes in a balanced, symmetrical distribution typical of granulomatous disease (either infectious or non-infectious); (**d**) a left upper lobe hypermetabolic nodule with ipsilateral hypermetabolic lymph node typical in appearance for the pattern of nodal spread of primary lung cancer (adenocarcinoma in this case).

**Figure 3 cancers-14-02689-f003:**
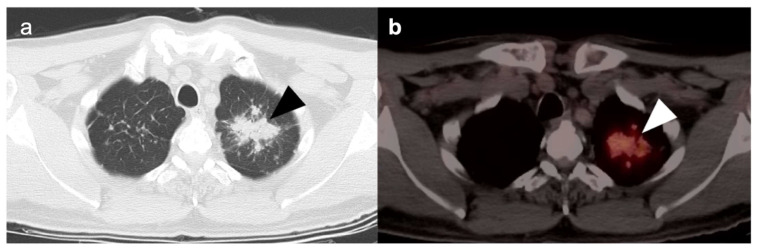
A 53-year-old male presenting with history of cough and shortness of breath was found to have an irregular opacity in the left upper lobe. (**a**) Axial CT (lung window) demonstrates an irregular mass lesion with speculated margins (**black arrowhead**); (**b**) axial FDG PET-CT demonstrated significantly increased metabolic activity correlating with the mass (**white arrowhead**). However, patient had long history of occupational exposure to silica as a worker in the mining industry and the finding of mass-like lesion was attributed to development of progressive massive fibrosis. This mass-like lesion was stable for multiple years of follow up. Pneumoconiosis can mimic primary lung malignancy and careful occupational history should be sought during evaluation of patients presenting with mass-like lesions in the lungs.

**Figure 4 cancers-14-02689-f004:**
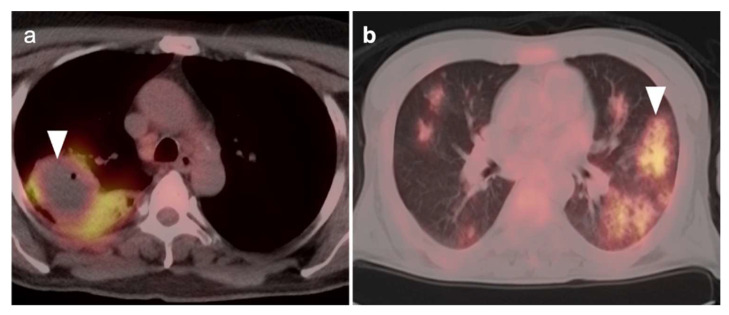
Pulmonary infection in two patients: (**a**) Case 1: Axial FDG PET-CT image demonstrates hypermetabolic consolidation in the right lower lobe with an associated peripherally hypermetabolic parenchymal fluid collection containing air consistent with pneumonia and a lung abscess (**white arrowhead**). There is also ipsilateral paratracheal mildly hypermetabolic reactive adenopathy; (**b**) Case 2: Multifocal ground glass consolidations (**white arrowhead**) in a patient with COVID-19 pneumonia. Both findings could mimic thoracic malignancy.

**Figure 5 cancers-14-02689-f005:**
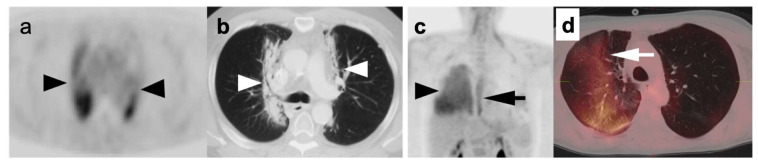
Post-radiation changes in the lungs in two patients: Case 1 (**a**,**b**): (**a**) Axial PET image demonstrates bilateral linear paramediastinal hypermetabolism (**black arrowheads**); (**b**) CT images of the chest (lung window) showing medial linear opacity (cicatricial atelectasis) and bronchiectasis (**white arrowheads**) resulting from chronic radiation fibrosis; Case 2 (**c**,**d**): (**c**) FDG PET scan shows diffuse hypermetabolism in the right lung (**black arrowhead**) and diffuse segmental hypermetabolism of the mid and distal esophagus (**black arrow**) on a MIP image; (**d**) The corresponding axial FDG PET-CT image of the chest shows corresponding hypermetabolic ground glass opacity in a geographic distribution in the right lung (**white arrow**) consistent with acute radiation pneumonitis. The patient had radiation of the esophagus that included a portion of the right lung for esophageal cancer.

**Figure 6 cancers-14-02689-f006:**
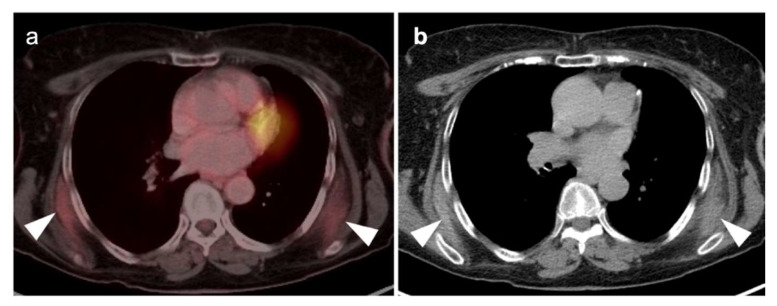
Elastofibroma dorsi: (**a**) Fused axial FDG PET-CT and (**b**) axial chest CT demonstrate mildly hypermetabolic diffuse soft tissue density between the latissimus dorsi and serratus muscles bilaterally (**white arrowheads**) consistent with elastofibroma dorsi.

**Figure 7 cancers-14-02689-f007:**
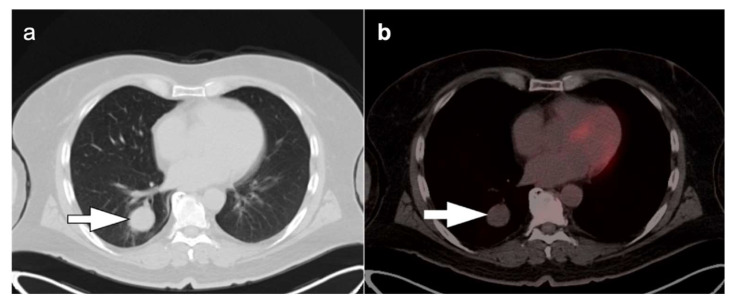
Bronchial carcinoid tumor. (**a**) Axial chest CT (lung windows) revealed a 3.0 cm, well-defined small mass with smooth margins in the right lower lobe (**white arrow**); (**b**) Corresponding FDG PET-CT showed minimal metabolic activity with an SUVmax of 1.2 (**white arrow**). Histopathology revealed a well-differentiated neuroendocrine tumor (typical bronchial carcinoid).

**Figure 8 cancers-14-02689-f008:**
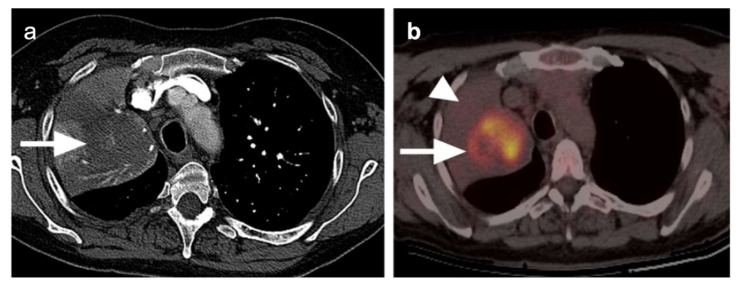
Non-small cell lung cancer. A 59-year-old male patient presented with large opacity in the right upper lobe. (**a**) Axial image from contrast enhanced CT study demonstrated a large irregularly hypoattenuating mass associated with lobar atelectasis (**white arrow**). However, it is difficult to delineate the mass lesion itself from the associated atelectasis or determine whether the mass extends to the pleural surface; (**b**) Fused axial FDG PET-CT images clearly differentiate the more central hypermetabolic mass lesion, with areas of internal photopenia representing areas of necrosis (**white arrow**). The area peripheral to this hypermetabolic mass is without significant metabolic activity and represents post-obstructive collapse of the pulmonary parenchyma distal to the mass (**white arrowhead**).

**Figure 9 cancers-14-02689-f009:**
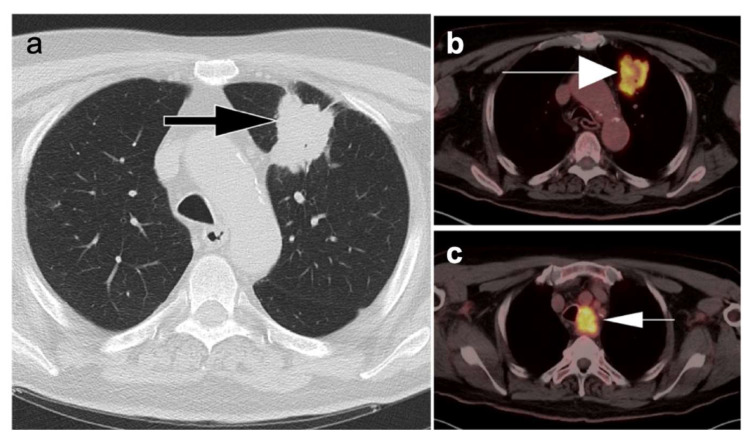
Patient presenting with a mass in the left upper lobe. (**a**) Axial chest CT image demonstrates an irregular shaped mass lesion in the left upper lobe (**black arrow**) with pleural tags; (**b**) Axial FDG PET-CT image demonstrates corresponding hypermetabolism in the left upper lobe lesion (**large white arrow**); (**c**) An enlarged hypermetabolic lymph node is identified in the ipsilateral left upper paratracheal region (**small arrow**). The lymph node was closely related to the lateral tracheal wall with possible invasion.

**Figure 10 cancers-14-02689-f010:**
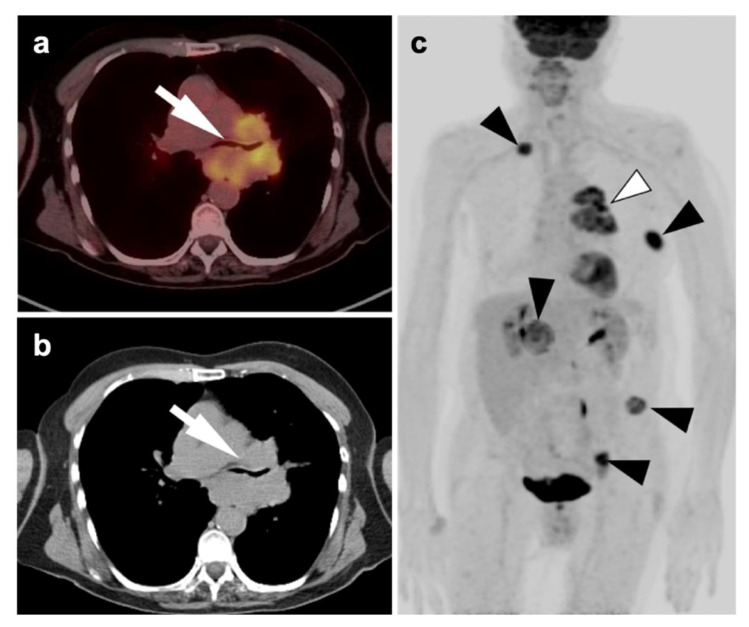
Patient with long history of cigarette smoking and history of weight loss, cough, and shortness of breath. (**a**) Fused axial FDG PET-CT axial and axial CT (**b**) images demonstrate a hypermetabolic mass involving the left hilar and perihilar region with narrowing of the left mainstem bronchus (**white arrows**); (**c**) whole-body MIP image demonstrates the left hilar hypermetabolic mass (**white arrowhead**), and multiple metastatic lesions (**black arrowheads**) in the right lower neck, left lateral chest wall (rib), right adrenal, left iliac wing, and a left pelvic sidewall node consistent with stage IV disease. Histopathology confirmed NSCLC.

**Figure 11 cancers-14-02689-f011:**
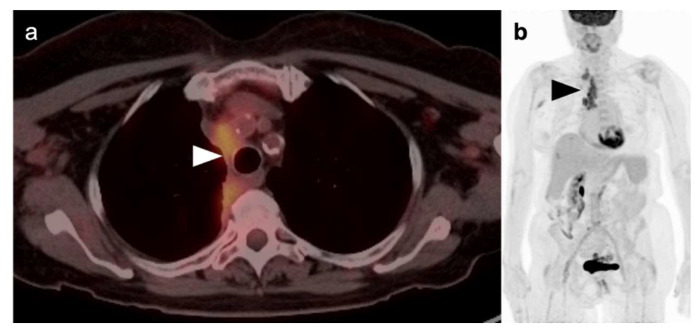
“Limited disease” small cell lung cancer. (**a**) Fused axial PET/CT image from a 73-year-old patient with biopsy-proven small cell lung cancer demonstrates an infiltrative hypermetabolic mass along the right paramediastinum (**white arrowhead**); (**b**) Whole-body FDG PET MIP image, shows that the hypermetabolic mass is extends superiorly to the high mediastinum and right medial supraclavicular region (**black arrowhead**). No other abnormal foci of uptake are identified. This patient is staged as “limited disease small cell lung cancer” (LD-SCLC).

**Figure 12 cancers-14-02689-f012:**
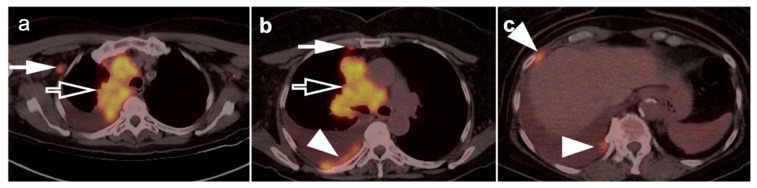
“Extensive disease” small cell lung cancer. (**a**–**c**) Fused axial FDG PET/CT images from a patient with biopsy-proven small cell lung cancer demonstrating an infiltrative, hypermetabolic mass in the right upper lobe extending into the mediastinum (**open arrow**). In addition, there are multiple metastatic lesions (**white arrows**), including a hypermetabolic lymph node in the right axilla (**a**), a right internal mammary node (**b**) and multiple focal tumor implants along the pleural surface (**white arrowheads**, **b**,**c**) associated with a large pleural effusion with metabolic activity within the fluid. Even though the tumor is confined to ipsilateral hemithorax, due to the presence of hypermetabolic pleural deposits, and axillary and internal mammary involved nodes, this patient is staged as “extensive disease small cell lung cancer” (ED-SCLC) rather than as “limited disease small cell lung cancer” (LD-SCLC).

**Figure 13 cancers-14-02689-f013:**
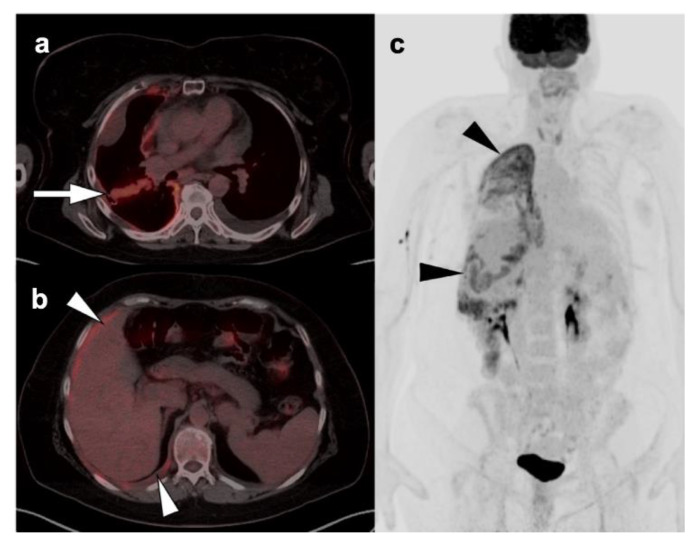
Malignant pleural mesothelioma. (**a**) Axial FDG PET-CT image shows a hypermetabolic rind of soft tissue within the pleura surrounding the right lung and tracking into the major fissure (**white arrow**); (**b**) Axial FDG PET-CT image shows a hypermetabolic tumor in the right deep costophrenic sulcus, which defines the lower margin of the pleural space (**white arrowheads**); (**c**) MIP image of the same FDG PET scan shows the distribution of hypermetabolic tumor in the pleural space (**black arrows**).

**Figure 14 cancers-14-02689-f014:**
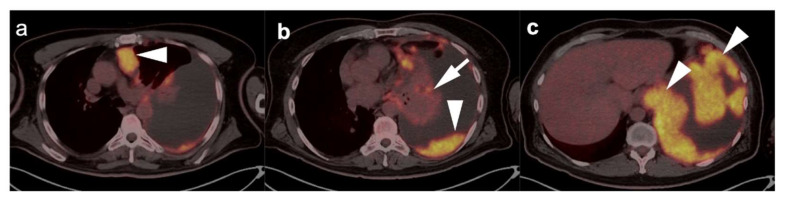
Malignant pleural mesothelioma (**a**–**c**). Bulky hypermetabolic soft tissue within the pleural space of the left lung (**white arrowheads**), associated with a large left pleural effusion and atelectatic collapse of the left lower lobe. Small focal areas of metabolic activity in the collapsed left lower lobe could be metastatic foci versus areas of superimposed inflammatory change (**white arrow**).

**Figure 15 cancers-14-02689-f015:**
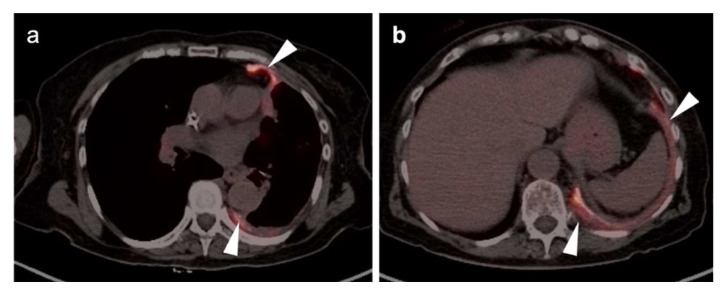
Reactive change secondary to talc pleurodesis: (**a**,**b**) The patient had undergone a talc pleurodesis several years earlier for recurrent spontaneous left pneumothorax. Areas of high attenuation with an associated thin ring of hypermetabolic soft tissue (**white arrowheads**) are consistent with chronic inflammatory host response to the presence of the talc. This could be confused with pleural calcified plaques and associated mesothelioma without the appropriate history.

**Figure 16 cancers-14-02689-f016:**
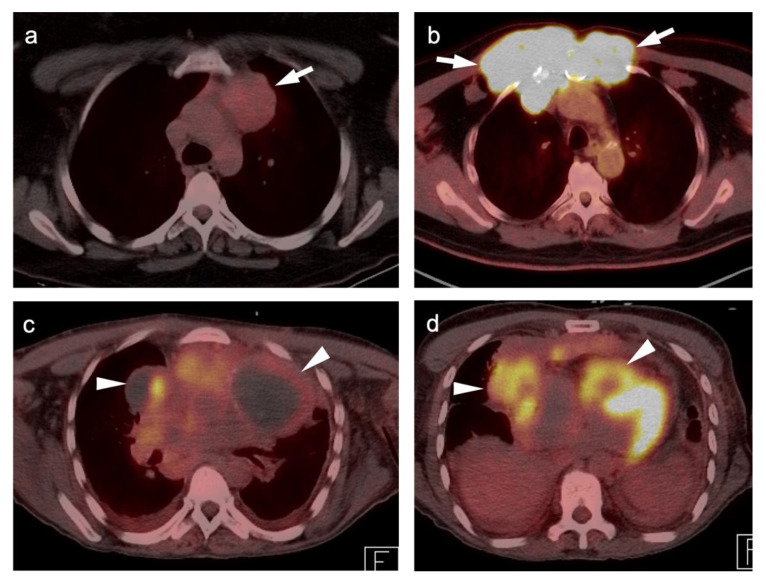
Three patients with primary thymic neoplasms. (**a**) Case 1: Thymoma. Smooth mild to moderate metabolically active mass (SUVmax is 5.2) in the left anterior mediastinum (**white arrow**) with many years of stability, biopsy-proven to be a thymoma; (**b**) Case 2: Thymic carcinoma. intensely hypermetabolic, invading the anterior chest wall (white arrows); (**c**,**d**) Case 3. Thymic carcinoma. (**c**) Axial FDG PET-CT image shows a complex anterior mediastinal mass with areas of cystic change (**white arrowheads**) and calcification (not well seen here); (**d**) Axial FDG PET-CT image at the level of the heart shows an intense metabolic portion of the mass (**white arrowheads**), biopsy proven to be a thymic carcinoma.

**Figure 17 cancers-14-02689-f017:**
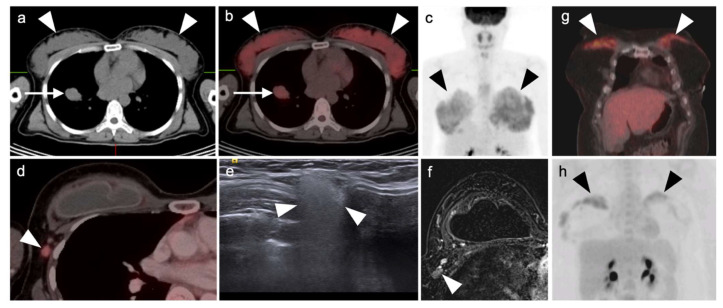
Non-malignant breast conditions. (**a**–**c**) Lactating breasts. (**a**) Lactating breast tissue is characterized by dense parenchymal tissue on axial CT (**white arrowheads**); (**b**) Axial FDG PET-CT shows increased metabolic activity in the dense lactating breast tissue (**white arrowheads**); (**c**) MIP FDG PET image shows bilateral dense hypermetabolic breast parenchyma typical with lactation. The right lung mass (**white arrow**) was a bronchial carcinoid tumor (successfully removed); The bronchial carcinoid is also metabolically active (**white arrow**); (**c**) FDG PET MIP image shows the hypermetabolic lactating breast tissue (**black arrowheads**); (**d**–**f**) Silicone granuloma. (**d**) Axial FDG PET-CT of the right breast shows a partially collapsed right breast implant with an enlarged hypermetabolic right axillary node (**white arrowhead**), a silicone granuloma. (**e**) Ultrasound of the right axillary node demonstrates a typical heterogeneous “snowstorm” sign of a silicone granuloma with posterior acoustic shadowing (**white arrowheads**). (**f**) Silicone specific STIR sequence MRI image demonstrates high signal intensity in lymph nodes due to presence of silicone (**white arrowhead**); (**g**,**h**) Fat necrosis. Coronal FDG PET-CT (**g**) and FDG PET MIP (**h**) images demonstrate hypermetabolic soft tissue within the superior margin of tram flap reconstruction following bilateral mastectomies, consistent with fat necrosis. This can remain hypermetabolic indefinitely.

**Figure 18 cancers-14-02689-f018:**
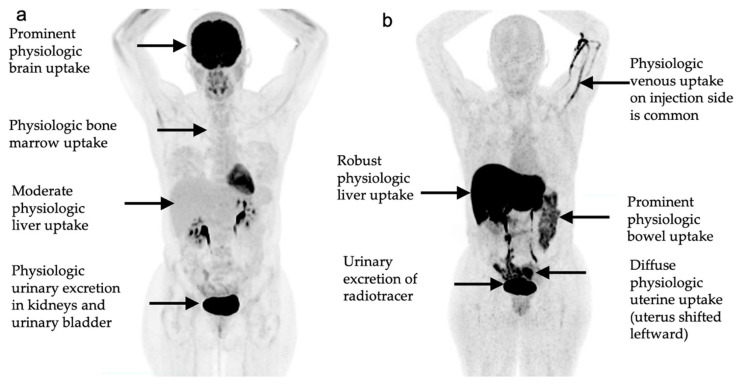
Normal physiologic biodistribution of PET radiotracer uptake for: (**a**) 18F-fluorodeoxyglucose (FDG); and (**b**) 18F-fluoroestradiol (FES).

**Figure 19 cancers-14-02689-f019:**
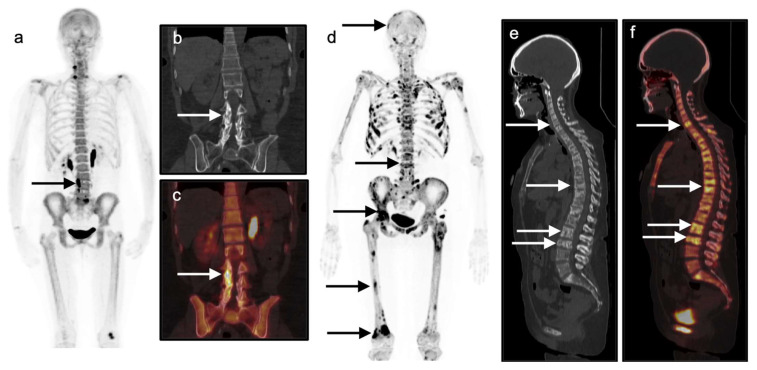
Comparison of degenerative (**a**–**c**) and malignant (**d**–**f**) uptake on ^18^F-sodium fluoride (NaF) PET imaging. (**a**–**c**) Degenerative disease. A 68-year-old female with biopsy-proven invasive ductal carcinoma of the breast presenting for evaluation of osseous metastatic disease with history of lumbar spine pain. NaF PET-CT imaging demonstrates multilevel uptake involving the right aspect of the L3-L5 vertebral bodies on the full body maximal intensity projection (MIP) image (**a**, **black arrow**). Corresponding coronal CT (**b**) and fused coronal PET/CT images (**c**) demonstrate that uptake corresponds with multilevel facet degenerative changes (**white arrows**); (**d**–**f**) Diffuse metastatic disease. In contrast, diffuse abnormal focal uptake is seen throughout the axial and appendicular skeleton on NaF PET-CT on a 49-year-old female with metastatic invasive ductal carcinoma with biopsy-proven osseous metastatic disease. MIP image (**d**) demonstrates >50 abnormal sites of focal uptake through the skeleton (**black arrows**, representative sites). Sagittal CT (**e**) and fused PET-CT (**f**) images demonstrate mixed lytic and sclerotic vertebral body lesions throughout the spine with corresponding abnormal focal radiotracer uptake consistent with multi-level osseous metastatic disease (**white arrows**).

**Figure 20 cancers-14-02689-f020:**
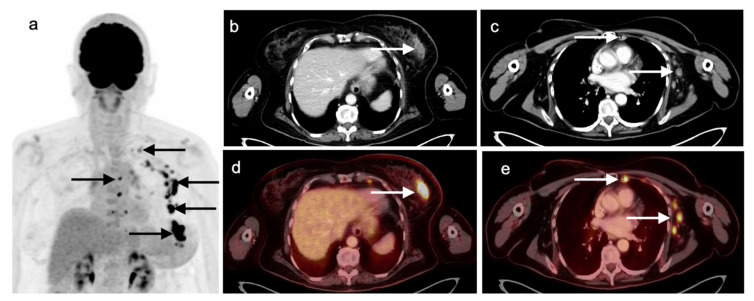
A 77-year-old female with left breast hormone-receptor-negative, HER2-positive invasive ductal carcinoma with biopsy-proven left axillary nodal metastatic disease. FDG PET-CT for initial staging demonstrates a hypermetabolic left breast mass (inferior-most black arrow on the maximal intensity projection image (**a**) and white arrows on axial CT (**b**) and fused FDG PET-CT images (**d**). Associated left axillary, left supraclavicular, left internal mammary, and mediastinal nodal metastatic disease is noted (superior **black arrows** on (**a**), **white arrows** at representative lesions on (**c**,**e**)).

**Figure 21 cancers-14-02689-f021:**
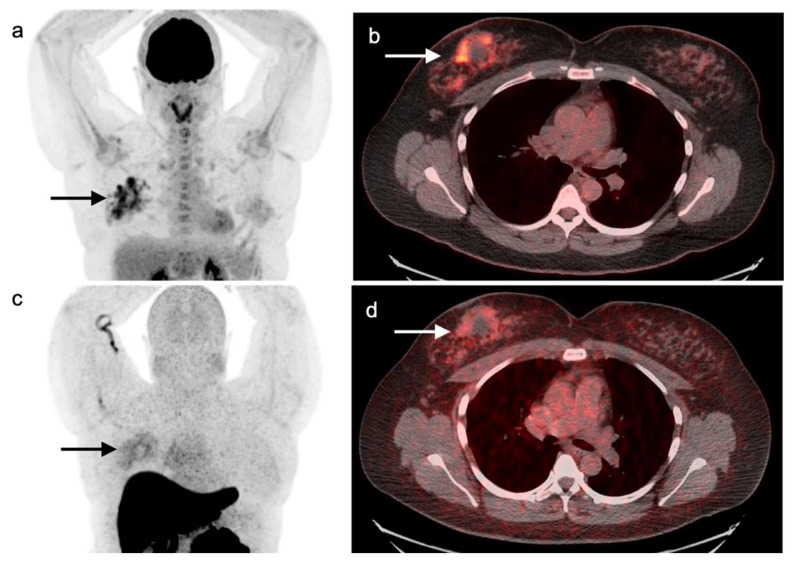
A 42-year-old female with biopsy-proven right breast ER+ invasive lobular carcinoma. Mastectomy demonstrated at least 9 cm of ILC diffusely in the right breast with negative sentinel lymph node biopsy. (**a**,**b**) FDG PET-CT imaging demonstrates abnormal asymmetric diffuse right breast uptake on MIP (**a**, **black arrow**) and fused axial PET-CT images (**b**, **white arrow**); (**c**,**d**) FES PET-CT imaging demonstrates abnormal diffuse asymmetric right breast uptake on maximal intensity projection (**c**, **black arrow**) and fused axial PET-CT images (**d**, **white arrow**).

**Figure 22 cancers-14-02689-f022:**
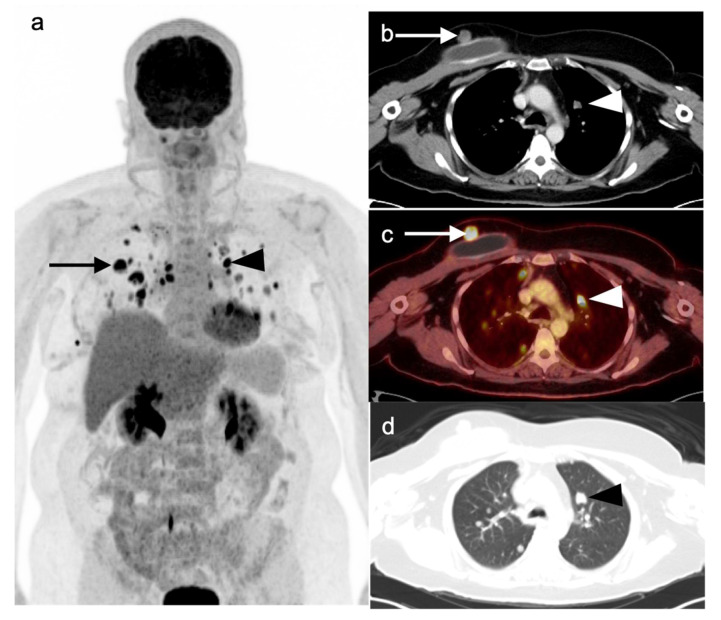
A 58-year-old female with histologically proven malignant phyllodes tumor of the right breast with biopsy-proven pulmonary metastases consistent with hematologic spread of malignancy to the lungs. MIP image (**a**) demonstrates the primary malignant phyllodes tumor of the superior right breast (**black arrow**) and abnormal focal uptake within greater than 10 bilateral lung metastases (representative black arrow). Axial CT with soft tissue windowing (**b**), fused PET/CT (**c**), and axial CT with lung windowing (**d**) show the right superior breast malignant phyllodes tumor (**white arrow** on **b**,**c**) superior to a right breast implant) and representative left upper lobe lung metastasis (**arrowheads** in **c**,**d**).

**Figure 23 cancers-14-02689-f023:**
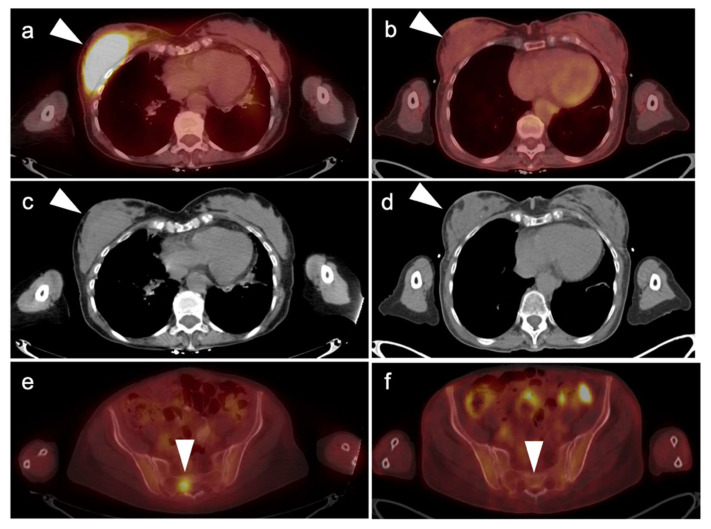
A 78-year-old female with diffuse large B-cell lymphoma presented with a palpable right breast mass, subsequently biopsied providing a diagnosis of diffuse large B-cell lymphoma. (**Left panel a**,**c**,**e**) Pretreatment images. FDG PET-CT demonstrates hypermetabolic right breast mass shown on fused axial PET-CT (**a**) and CT (**c**) images (**white arrowheads**); (**e**) A hypermetabolic bone lesion was identified in the sacrum consistent with stage 4 disease, shown on a fused axial PET/CT image (**c**, **white arrowhead**); (**right panel**, **b**,**d**,**f**) follow-up FDG PET-CT performed for therapeutic response assessment following chemotherapy demonstrated a complete metabolic response to therapy. Following 2 cycles of therapy, repeat FDG PET-CT was performed demonstrating a complete metabolic response to therapy as shown on axial fused PET-CT (**b**,**f**) and CT (**d**) images (**white arrowheads**).

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
