# Peer review of "PET-CT in Clinical Adult Oncology: II. Primary Thoracic and Breast Malignancies"

_cancers, 2022, doi:10.3390/cancers14112689_

Round 1
Reviewer 1 Report
Abstract: well written
Introduction: Is too short. I don't see any references in the text. Write also, the aim of this review and what do you intend to present.
Table 1: We already know this table. No new information has been brought up. This table should be removed.
Figure 17 is missing.
Conclusion is way too long. Needs to be re-aranged.
Reviewer 2 Report
In this review article the authors discuss evaluation of primary thoracic and breast malignancies
with PET-CT
Strengths of the article:
- The review is expensive and of interest to clinicians, particularly trainees in radiology and oncology
- The review provides information on several new imaging agents, including FES PET/CT imaging in ER+ breast cancers.
- The article is well written.
Concerns about the study:
- The review mentions that PET/CT can be used for evaluation of therapy response. However, not much detail is provided regarding therapy response approaches in PET and its advantages and disadvantages compared with CT and MRI therapy assesdment. Furthermore, it would be helpful to learn more about the pitfalls of assessment of therapy response with PET to novel systemic therapy agents such as immunotherapy.
- It would be helpful to provide more specific examples of false positives and false negatives, particularly with the novel agent FES (fulvestrant and tomoxifen interactions, metastatic disease in the liver).
- It would be helpful to see a little more statistical data comparing various modalities (for example in diagnosis with ROC AUCs and outcomes with hazard ratios etc) than what is currently reported in the article..
Reviewer 3 Report
K. A. Morton et al., written review article PET-CT in Clinical Adult Oncology: II. Primary Thoracic and Breast Malignancies, is interesting and can be consider for publication after revised the below comments.
- Abstract or Introduction:- author has to cite the reference of I st review article of this series, so the readers can follow it easily.
- missing interesting review article of FDG for Breast cancer study :- page no 19 2nd sentence " PET-CT for breast.....FDG. should be address by recent review article by O. Prante et al., Pharmaceuticals, 2021, 14, 1175.
- Page no 20, line 675, author missed a image of Figure 17. the review article will be useful for oncology researcher and medicine field.
